# Genome-Wide Identification and Expression Analysis of the *CAMTA* Gene Family in Roses (*Rosa chinensis* Jacq.)

**DOI:** 10.3390/plants14010070

**Published:** 2024-12-29

**Authors:** Wanyi Su, Yuzheng Deng, Xuejuan Pan, Ailing Li, Yongjie Zhu, Jitao Zhang, Siting Lu, Weibiao Liao

**Affiliations:** College of Horticulture, Gansu Agricultural University, Lanzhou 730070, China; s18294911904@163.com (W.S.); dengyz0830@126.com (Y.D.); panxj@st.gsau.edu.cn (X.P.); 17899315228@163.com (A.L.); 18793284582@163.com (Y.Z.); zjt117769@163.com (J.Z.); m17339887987@163.com (S.L.)

**Keywords:** roses, *CAMTA* gene family, phylogenetic relationships, gene expression, phytohormone-signaling response

## Abstract

Calmodulin-binding transcription activator (*CAMTA*), as one of the transcription factors, is involved in performing important functions in modulating plant stress responses and development in a Ca^2+^/CaM-driven modus. However, genome-scale analysis of *CAMTA* has not been systemically investigated in roses. Rose (*Rosa chinensis* Jacq.) *CAMTA* gene family members were identified and bioinformatically analyzed to investigate their expression characteristics in plant hormonal responses. The results show that a total of five rose *CAMTA* genes were identified. Chromosomal localization shows that the *RcCAMTA* gene members were located on chromosomes 2, 4, and 7. Physicochemical property analysis shows that its CDS sequence length ranges from 500 to 1070 bp, the molecular weight ranges from 55,531.60 to 120,252.98 Da, and the isoelectric point is from 5.04 to 8.54. Phylogenetic analysis shows that rose *CAMTA* genes are classified into three subfamilies. Conservative motif analysis reveals the presence of motif 1, motif 3, motif 5, motif 7, and motif 10 in all the *RcCAMTA* genes. The cis-acting element prediction results show that the rose *CAMTA* gene family contains phytohormone-signaling response elements, abiotic stress responses, light responses, and other elements, most of which are hormone-signaling response elements. From the expression levels of *RcCAMTA* genes, the *CAMTA* family’s genes in roses have different spatial expression patterns in different tissues. The qRT-PCR analysis showed that all five rose *CAMTA* genes responded to salicylic acid (SA). *RcCAMTA3* was significantly induced by abscisic acid (ABA), and *RcCAMTA2* was significantly induced by 1H-indole-3-acetic acid (IAA) and methyl jasmonate (MeJA). Thus, we provide a basic reference for further studies about the functions of CAMTA proteins in plants.

## 1. Introduction

Calcium (Ca^2+^) ions are involved in many cellular signaling pathways as prevalent secondary messengers in eukaryotes [1]. Ca^2+^-mediated signaling plays a key role in the transmission of signals generated by different stimuli. Thus, it mediates various stress responses in plants [2,3]. CaM can bind to Ca^2+^ as flexible Ca^2+^/CaM structural proteins, and Ca^2+^ in structural proteins can interact with many proteins, allowing for CaM in structural proteins to regulate protein targets in many different signaling pathways [4,5]. To date, gene families such as *CAMTA*, *MYB*, *WRKY*, *NAC*, and *bZIP* have been identified as Ca^2 +^/CAM-binding proteins [6,7,8,9,10].

Calmodulin-binding transcription activator (*CAMTA*), as one of the transcription factors, is involved in performing important functions in modulating plant stress responses and development in a Ca^2+^/CaM-driven modus [11]. CAMTA proteins consist of multiple predicted functional domains, and these functional domains include (1) a CG-1 domain, containing the predicted bipartite nuclear localization signal (NLS, a specific amino acid sequence that indicates the localization of proteins in cells); (2) a transcription factor immunoglobulin-binding domain (TIG), which was reported to be implicated in nonspecific DNA contacts in various transcription factors (TFs) and involved in protein dimerization; (3) ankyrin (ANK) repeats, known to be involved in nonspecific protein–protein interactions and present in a large number of functionally diverse proteins; and (4) a variable number of 10 motifs, known as calmodulin-binding sites, localized in the C-terminal part of CAMTA proteins [12,13]. To date, the protein domains and gene structures of CAMTA proteins have been reported in detail in Arabidopsis (*Arabidopsis thaliana* L.) [12], tomato (*Solanum lycopersicum* L.) [14], rice *(Oryza sativa* L.) [15], maize *(Zea mays* L.) [16], cotton (*Gossypium hirsutum* L.) [17], tobacco (*Nicotiana tabacum* L.) [18], alfalfa (*Medicago sativa* L.) [19], grapes *(Vitis vinifera)* [20], kidney bean *(Phaseolus vulgaris* L.) [21], and soya bean *(Glycine max* L.) [22].

The *CAMTA* family regulates gene expression by binding cis-elements in the promoter regions of target genes. The cis-element binding to *CAMTA* genes was first discovered in Arabidopsis, with the sequence (G/A/C) CGCG (C/G/T). The recognition sequence in rice *CAMTA* genes is (A/C) CGTGT, which is different from that in *A*. *thaliana*. Among them, (A/C) CGTGT also contains abscisic acid (ABA)-responsive elements (ABRE: ACGTGT) [23,24]. *CAMTA* genes respond to 1H-indole-3-acetic acid (IAA), ethylene (ETH), ABA, and salicylic acid (SA) [22,25]. In *A*. *thaliana*, *AtCAMTA1* plays a role in the IAA-signaling response and responds to drought stress by producing ABA [26]. In citrus (*Citrus reticulata* Blanco), the responses to and regulatory effects of *CAMTA* gene family members on hormones are equally complex and diverse. The citrus *CAMTA* gene responds to various hormonal treatments, such as SA and ETH, and affects the growth, development, and stress resistance of citrus by regulating the expressions of related genes [27]. Members of the *CAMTA* gene family in rice can respond to hormonal treatments, such as IAA and cytokinin (CTK), and affect growth and development in rice by regulating the expressions of related genes [28]. It is worth noting that members of the *CAMTA* gene family may have different responses to and regulatory effects on hormones in different species. These differences may stem from genetic differences between species, differences in ecological environments, and the complexity of hormone-signaling networks [29]. Therefore, when studying the effects of the *CAMTA* gene family on hormones, it is necessary to fully consider the differences between species and the complexity of hormone-signaling networks [18].

Roses (*Rosa chinensis* Jacq.), among the most important ornamental flowers, not only have extremely high esthetic value but also possess substantial economic significance [30]. In our present study, we identified five *CAMTA* genes in the rose “Movie Star” and employed a systematic bioinformatics analysis to study *RcCAMTA* gene family members, using genomic sequencing. For *RcCAMTA* member identification, we analyzed the gene structure, motif composition, chromosomal location, and synteny of roses. In addition, *RcCAMTA* gene expressions in stems, leaves, petals, pistils, stamens, calyxes, receptacles, and prickles were analyzed based on transcriptomic data, and *RcCAMTA* genes under different hormonal treatments were selected for expression analysis via quantitative real-time polymerase chain reaction (qRT-PCR). This study provides details about the evolutionary and functional characteristics of the *CAMTA* gene family in roses and lays a foundation for the future functional characterization of *CAMTA* genes.

## 2. Results

### 2.1. Genome-Wide Identification and Chromosomal Location of the RcCAMTA Gene Family

The genome-wide search resulted in the identification of five *CAMTA* genes in the rose genome, which were named *RcCAMTA1* to *RcCAMTA5* as per their distribution in the rose chromosomes (Table 1; Figure 1A). The five rose *CAMTA* family members are unevenly distributed in chromosomes 2, 4, and 7. Among them, chromosomes 2, 4, and 7 possess 2, 2, and 1 *CAMTA* members, respectively (Figure 1A). As shown in Figure 1B, *RcCAMTA1* contains four exons and three introns, *RcCAMTA2* and *RcCAMTA3* each contain thirteen exons and twelve introns, and *RcCAMTA4* and *RcCAMTA5* each contain twelve exons and eleven introns. The number of amino acids in the encoded protein ranges from 500 to 1070 aa (Table 2). *RcCAMTA1* has the smallest number of amino acids and the lowest molecular weight; *RcCAMTA2* has the largest number of amino acids and the highest molecular weight. Furthermore, the isoelectric point (pI) changes from 5.04 (*RcCAMTA1*) to 8.54 (*RcCAMTA5*). The instability index changes from 40.26 (*RcCAMTA1*) to 49.11 (*RcCAMTA4*), with no instability index less than 40, indicating that they are all unstable proteins. The aliphatic index changes from 74.03 (*RcCAMTA2*) to 81.67 (*RcCAMTA3*). The total mean values of hydrophilicity are all negative, indicating that the *CAMTA* family members are less hydrophilic. In addition, subcellular localization prediction suggests that the RcCAMTA1 protein is localized in the endoplasmic reticulum and the nucleus, and the RcCAMTA2, RcCAMTA3, RcCAMTA4, and RcCAMTA5 proteins are localized in the nucleus (Table 1).

### 2.2. Predictive Analysis of the Secondary and Tertiary Structure of RcCAMTAs

The secondary structure of the CAMTA gene protein including the proportion and distribution of α-helix, β-angle, and random curling was predicted by analyzing the hydrogen bond formation ability and hydrophobicity among amino acid residues. The results show that the proportion of α-helix in the RcCAMTA proteins ranges from 32.80% to 44.34%, the chain extension structure ranges from 3.33% to 5.43%, the irregular curling ranges from 56.05% to 62.99%, and there is no β-angle (Table 2). Therefore, it is speculated that certain peptide chains do not form β-angle structures, or the proportion of beta angles in the structure is so low that it can be considered “0” or non-existent. To determine the accuracy of the protein secondary structure predictions, homology modeling was performed for five members of the rose *CAMTA* family. A three-dimensional structure model of the target protein was constructed by searching for a template structure, performing sequence alignment, and then constructing and optimizing the model. As shown in Figure 2, the predicted protein tertiary structure was basically consistent with the secondary structure.

### 2.3. Multiple Sequence Alignment and Phylogenetic Tree of RcCAMTAs

The results show that the CAMTA proteins have conserved structural domains, including four from the N-terminus to the C-terminus CG-1, IPT, ANK, and IQ (Figure 3A). We used the maximum likelihood method to construct a phylogenetic tree. It includes five rose CAMTA proteins, six Arabidopsis CAMTA proteins, 18 rice CAMTA proteins, and seven maize CAMTA proteins (Figure 3B). Based on sequence similarity, the 36 CAMTA proteins were divided into four subfamilies. Group 1 has two CAMTA proteins from Arabidopsis; group 2 contains six and three CAMTA proteins from rice and maize, respectively; group 3 contains five and two CAMTA proteins from rice and maize, respectively; and group 4 contains five, four, seven, and two CAMTA proteins from rose, Arabidopsis, rice, and maize, respectively. Among them, the rose CAMTA family members are uniquely distributed in group 4. Only one pair of genes, *RcCAMTA1* and *RcCAMTA2*, are in the same subgroup and the same branch, while *RcCAMTA3*, *RcCAMTA4,* and *RcCAMTA5* are in the same subgroup but different branches. The latter three proteins are close in affinity and hypothesized to have similar protein functions.

### 2.4. Conserved Motifs of Rose CAMTA Proteins

In the rose CAMTA family, the maximum value was set to 10 conserved motifs (Figure 4). The length of these motifs ranges from nine to fifty amino acids. Using TBtools-II software, we found that rose CAMTA proteins belonging to the same subfamily in the evolutionary tree contain similar or identical Motif compositions. Most rose CAMTA proteins contain 10 common motifs in the order of motif 6, motif 2, motif 8, motif 7, motif 1, motif 3, motif 5, motif 9, motif 4, and motif 10, while some CAMTA proteins contain only part of the motifs, and some parts of the motifs are missing. For example, RcCAMTA1 contains only motifs 7, 1, 3, 5, and 10 (Figure 4A). The sequence information of the characterized conserved motifs is shown in Appendix A, where the amino acid sequences of the different conserved motifs are indicated by a stack of letters at each position (Figure 4B).

### 2.5. Interaction Network of Rose CAMTA Proteins

The protein interaction network of the rose CAMTA family consists of 50 proteins, which form 70 protein interactions (Figure 5). Among them, the RcCAMTA3 protein is the core protein, which interacts with up to 28 other proteins; the RcCAMTA4 protein interacts with 21 other proteins; the RcCAMTA5 protein interacts with 18 other proteins. It can be seen that in the protein interaction network, the RcCAMTA3, RcCAMTA4, and RcCAMTA5 proteins act as central nodes that interact with multiple other proteins, thus allowing for the coordination of multiple biological processes. RcCAMTA2 consists of only three interactions, and RcCAMTA1 does not have any interactions (Figure 5). It can be inferred that RcCAMTA1 may act as a structural protein, such as collagen or keratin, etc., and does not need to interact with other proteins to perform its functions.

### 2.6. Collinearity Analysis of Rose CAMTA Family Genes 

To further investigate the evolution of *CAMTA* family members in rose, genome collinearity analysis maps of Arabidopsis and rose, as well as apple (*Malus pumila* Mill.) and rose, were constructed based on the results of collinearity analysis between the species. Among them, the number of homologous gene pairs of *CAMTA* family members is two between rose and Arabidopsis (Figure 6A) and four between rose and apple (Figure 6B). The results suggest that rose and apple have more homologous genes and that the divergence between rose and apple occurred after their common dicot ancestor diverged from the dicot ancestor of Arabidopsis.

### 2.7. Promoter Cis-Acting Element Analysis of RcCAMTA

The predicted cis-acting elements were classified as hormone signal response, abiotic stress response, or light signal response, according to their functions and actions (Appendix A). The *RcCAMTA* gene contains 14 cis-elements that respond to hormones and stresses (Figure 7A). Several elements that respond to abiotic stress, namely the LTR, ARE, MBS, and GC motif repeats, were identified. In terms of light signal response, circadian, G-box, O^2^-site, and P-box were identified. In addition, many *RcCAMTA* genes contain hormone signaling response elements, such as the TGA-element, CGTCA motif, TCA-element, and ABRE. Elements such as the LTR, ARE, G-box, ABRE, TCA-element, and CGTCA motif are mainly distributed in *RcCAMTA1* (Figure 7B). The LTR, ARE, GC-motif, MBS, G-box, O^2^-site, ABRE, TGA-element, TCA-element, and CGTCA-motif are mainly distributed in *RcCAMTA2*. The LTR, ARE, MBS, G-box, P-box, ABRE, TGA-element, and CGTCA motif are mainly distributed in *RcCAMTA3*. Elements such as the ARE, G-box, P-box, ABRE, TCA-element, and CGTCA motif are mainly distributed in *RcCAMTA4*. Elements such as the LTR, ARE, MBS, ABRE, TGA-element, and TCA-element are mainly distributed in *RcCAMTA5* (Figure 7B). Our examination of cis-regulatory regions revealed that a large fraction of the *RcCAMTA* gene may have a significant impact on hormone response.

The results of the cis-acting element analysis indicate that the *RcCAMTA* genes respond to most hormone response elements, so we investigated the effects of various hormones (salicylic acid, abscisic acid, methyl jasmonate, and 1H-indole-3-acetic acid) on postharvest preservation in cut rose. According to our observations, the cut roses began to show some degree of wilting at 5 days. Therefore, we performed an analysis of cut rose flower phenotypes at 5 days (Figure 8A). It can be clearly seen that the petals showed obvious wilting under the MeJA treatment, and the calyx showed a certain degree of chlorosis, indicating the inhibitory role of MeJA in the freshness of cut roses. Compared with the control, the peripheral petals of the cut roses treated with IAA and ABA showed only a slight degree of water loss, but the chlorosis in the calyx was more pronounced. In contrast, compared with the control, the cut flowers treated with SA were bright, firm and more vigorous (Figure 8A).

The vase life of the cut roses under the IAA and ABA treatments was shorter than that of the control (Figure 8B). The SA treatment resulted in the longest vase life. The MeJA treatment resulted in the shortest vase life among all treatments (Figure 8B). As shown in Figure 8C, the IAA, ABA, and MeJA treatments resulted in the largest flower diameters on day 4; while the control and SA treatment resulted in the maximum flower diameters on day 6.

### 2.8. Expression Analysis of RcCAMTAs in Different Tissues

To study the spatial expression pattern of *RcCAMTAs*, we analyzed the expression levels of *RcCAMTA* genes in eight tissues, including the stems, leaves, petals, pistils, stamens, calyces, receptacles, and thorns (Figure 9). The results show that *RcCAMTAs* were expressed in all tissues at different levels. In the stamens, the highest levels of *RcCAMTA4* were expressed and the lowest levels of *RcCAMTA5* were expressed. In the pistils, the expression levels of *RcCAMTA1*, *RcCAMTA2*, and *RcCAMTA4* were highly expressed. In the petals, the expression level of *RcCAMTA2* was the highest. *RcCAMTA2* and *RcCAMTA5* had low expression levels in the receptacles and stems, while the other *RcCAMTAs* were highly expressed. *RcCAMTA1* and *RcCAMTA3* were expressed at high levels in the calyces. *RcCAMTA1* and *RcCAMTA4* were highly expressed in the leaves, and *RcCAMTA5* was expressed at a low level. *RcCAMTA3* was expressed at the highest level in the thorns (Figure 9). The expression levels of the *RcCAMTAs* suggest that *CAMTA* family genes may have different spatial expression patterns in different tissues of rose.

### 2.9. Expression Analysis of RcCAMTA Genes Under Different Treatments

The expression patterns of rose *CAMTA* genes were analyzed under different hormone treatments (IAA, ABA, SA, and MeJA) (Figure 10). Under the IAA treatment, there were significant differences in the expression patterns of the *RcCAMTAs* over time. *RcCAMTA1*, *RcCAMTA4*, and *RcCAMTA5* were down-regulated after 6 h of the IAA treatment compared to the control. *RcCAMTA2* and *RcCAMTA3* were up-regulated at the beginning of the treatment and reached their maximum at 12 and 6 h, respectively, with expression levels 1.91 and 1.47 times higher than the control (Figure 10).

After the ABA treatment, the expressions of *RcCAMTA1* and *RcCAMTA2* were up-regulated. The expression of *RcCAMTA1* and *RcCAMTA2* reached their maximum at 24 and 48 h, respectively, with levels 1.71-fold and 2.19-fold higher than the control, respectively. The expression of *RcCAMTA3* was significantly up-regulated under the ABA treatment at 6 and 9 h, with the maximum value reached at 6 h, with a level 3.11-fold higher than the control. The expression of *RcCAMTA4* was significantly lower than the control at 6 h and then began to increase until it was significantly higher than the control at 48 h. There was no significant difference for the rest of the treatment time. The expression level of *RcCAMTA5* was lower with the ABA treatment than in the control for the entire duration (Figure 10).

All *CAMTA* genes significantly responded to the SA treatment, but their expression patterns were different. The relative expressions of *RcCAMTA1*, *RcCAMTA2*, *RcCAMTA3,* and *RcCAMTA5* were all significantly up-regulated after 6 h of SA treatment, where they reached their maximum at 12 h, with levels 3.56, 3.62, 2.47, and 2.5-fold higher than the control, respectively. The transcript abundance of *RcCAMTA4* was significantly up-regulated at 6, 12, 24, and 96 h of the SA treatment, with the maximum expression reached at 12 h, with a level 3.22 times that of the control (Figure 10).

MeJA up-regulated the expressions of *RcCAMTA1*, *RcCAMTA2,* and *RcCAMTA3*. Their expressions peaked at 24, 12, and 24 h, respectively, with levels 1.58-fold, 2.8-fold, and 2-fold higher than the control. The expression of *RcCAMTA4* was significantly down-regulated throughout the MeJA treatment period. The expression of *RcCAMTA5* was significantly higher at 6 h with the MeJA treatment compared to the control (Figure 10).

## 3. Discussion

Bioinformatics tools and publicly released genomics data have led to the identification of numerous plant gene families, especially in model plants such as Arabidopsis, tobacco, and rice. So far, the CAMTA transcription factor family has been widely identified in various plants [15]. For example, six *CAMTA* members (*AtCAMTA1*~*AtCAMTA6*) were identified in *A. thaliana* L. [12], seven *CAMTA* members (*MtCAMTA1*~*MtCAMTA17*) were identified in alfalfa (*Medicago truncatula*) [19], fifteen *CAMTA* members (*TaCAMTA1*~*TaCAMTA15*) were identified in wheat (*Triticum aestivum* L.) [31], seven *CAMTA* members (*OsCAMTA1*~*OsCAMTA7*) were identified in rice [28], eleven *CAMTA* members (*PeCAMTA1*~*PeCAMTA11*) were identified in Moso bamboo *(Phyllostachys edulis)* [32], and eighteen *CAMTA* members (*BnCAMTA1*~*BnCAMTA18*) were identified in rapeseed (*Brassica napus* L) [33]. These findings collectively reveal the universality and diversity of *CAMTA* transcription factors in different plant species [18]. However, genome-wide identification and annotation of *CAMTA* genes have not been reported in cut rose. In this study, five *CAMTA* family genes were identified in the *R. chinensis* genome (Table 2). Among them, *RcCAMTA1* has only half the amino acid length of the other members and lacks many structural domains. Research has found that, among the members of the *CAMTA* family in apples, *MdCAMTA6* and *MdCAMT9* also exhibit a phenomenon where their amino acid length is half that of the other members, and they also lack many structural domains [34]. *RcCAMTA* gene members are located on chromosomes 2, 4, and 7 (Figure 1). The homology among different *CAMTA* genes indicates that during the evolution of rose, members of this gene family have experienced duplication or recombination events. It is worth noting that on chromosome 2, *RcCAMTA1* and *RcCAMTA2* are closely adjacent, forming a gene cluster. This arrangement suggests that these two genes may coencode proteins and jointly participate in regulating specific biological processes. In addition, the *CAMTA* gene also exhibits tandem duplication in alfalfa, with *MsCAMTA15*, *MsCAMTA16*, and *MsCAMTA17* forming a tandem duplication gene cluster on the Chr8.3 chromosome [35].

The similar conserved sequences and gene structures among members of the *CAMTA* gene family indicate that the biological functions of these genes are usually the same within a family. For example, the six *NTR1* homologs found in Arabidopsis all exhibit a common structural feature: the N-terminus contains a DNA-binding region (CGCG domain), while the C-terminus contains a calmodulin (CaM)-binding domain [36]. The role of Ca^2+^/CaM may involve regulating interactions with other proteins or modulating transcriptional activation processes. Furthermore, comparative analysis of conserved domains revealed some important characteristics of the *RcCAMTA* gene. Specifically, four *RcCAMTA* genes carry CG-1 domains, two *RcCAMTA* genes carry IPT domains, and five *RcCAMTA* genes have ANK (anchor protein repeat) domains. Similarly, four *RcCAMTA* genes have IQ domains (Figure 3A). These observations strongly suggest that the conserved motifs in the *CAMTA* family have maintained a high degree of consistency throughout their evolutionary history, which has helped maintain their fundamental functions across different species. We used the maximum likelihood method to construct a phylogenetic tree. According to the sequence similarity, the 36 CAMTA proteins are divided into four subfamilies. Among them, the *RcCAMTA* family members are only distributed in group 4 (Figure 3B). Xie et al. [37] also observed a similar phenomenon in the evolutionary tree analysis of citrus SAUR transcription factors, where the 12 *SAUR* members of citrus are only distributed in B7.

The adaptive response of plants to hormone stimulation involves a series of complex biological processes, including the perception of hormone signals, intracellular signal transduction, the activation of cis-acting elements, and regulation of the expression of a large number of related genes [38]. This series of events is achieved by activating specific hormone signaling pathways, which in turn affect plant growth, development, and other physiological functions. Transcription factors play a crucial role in this process, as they can recognize and bind to specific cis-acting elements in the promoter regions of target genes, thereby regulating their transcriptional activity [39]. *CAMTA* regulates gene expression by binding cis-elements in the promoter region of target genes. The promoters of the rose *CAMTA* gene family members contain multiple cis-acting elements related to hormone signal response, such as the IAA-responsive element, the TGA-element, the ABA-responsive element ABRE, the SA-responsive element, the TCA-element, the MeJA-responsive element, and the CGTCA motif (Appendix A). Based on this, we used different exogenous hormones to study the vase life of roses and found that the MeJA, IAA, and ABA treatments decreased the vase life, while the SA treatment increased it (Figure 7). It is speculated that *RcCAMTAs* may play an important role in regulating hormone response. The role of *CAMTA* genes in phytohormone response has also been studied in other species. For example, in *A. thaliana* L., *AtCAMTA1* plays a role in the IAA signaling response and also responds to drought stress by producing ABA [26]. In citrus (*Citrus reticulata Blanco*), the response and regulatory effects of the *CAMTA* gene family members on hormones are equally complex and diverse. The citrus *CAMTA* gene responds to various hormone treatments, such as SA and ETH, and affects the growth, development, and stress resistance of citrus by regulating the expressions of related genes [27]. Members of the *CAMTA* gene family in rice can respond to hormone treatments, such as IAA and cytokinin (CTK), and affect the growth and development of rice by regulating the expressions of related genes [28]. It is worth noting that members of the *CAMTA* gene family may have different responses and regulatory effects on hormones in different species. These differences may stem from the genetic differences among species, differences in ecological environments, and the complexity of hormone signaling networks [29]. Therefore, when studying the roles of the *CAMTA* gene family in hormone signaling, it is necessary to fully consider the differences among species and the complexity of hormone signaling networks.

Tissue expression pattern analysis revealed that *RcCAMTA* genes were expressed in the stems, leaves, flowers, pistils, stamens, calyces, receptacles, and thorns (Figure 8). This indicates that these genes play important roles in the different tissues of roses. As research continues to advance, it has been found that CAMTA transcription factors in different plants have different responses to IAA, ABA, SA, and MeJA. The expression patterns of rose *CAMTA* genes under different hormone treatments were analyzed (Figure 9). The expressions of *RcCAMTA4* and *RcCAMTA5* were down-regulated after 6 h of the IAA treatment and then remained relatively stable. It is worth noting that similar results were found in citrus, where the expression level of *CitCAMTA3* suddenly decreased after 3 h of IAA stimulation, and then remained mostly stable [27]. Thus, *RcCAMTA4, RcCAMTA5,* and *CitCAMTA3* might respond rapidly to the IAA treatment. Our results also show that the ABA treatment rapidly up-regulated the expression levels of *RcCAMTA3*. It is worth noting that similar results were found in foxtail millet (*Setaria italica* L.). where the expressions of *SiSHMT1*, *SiSHMT2*, and *RcCAMTA3* were significantly up-regulated at 6 h after an ABA treatment [40]. All *RcCAMTA* genes significantly responded to the SA treatment. In citrus, eight out of nine members of *CitCAMTA* were significantly up-regulated by SA treatment [27]. The expression of *RcCAMTA4* was significantly down-regulated throughout the MeJA treatment period (Figure 9). In the research results of Zhang Jing et al., the expression levels of *CitCAMTA3* and *CitCAMTA8* were inhibited under the MeJA treatment [27]. Our study suggests that the *RcCAMTA* gene family plays a crucial role in the hormone stress response, but further studies are needed to elucidate the functional significance of the CAMTA gene family in rose.

## 4. Materials and Methods

### 4.1. Identification and Sequence Analysis of Rosa Chinensis CAMTA Genes

The *R*. *chinensis* genome file was downloaded from the NCBI website (https://www.NCBI.nlm.nih.gov/, accessed on 15 June 2024). Then, we downloaded the CAMTA DNA-binding domain (Pfam: PF03859, PF00612 and PF-00023) from the Pfam online database (http://pfam.xfam.org, accessed on 17 June 2024). Similarly, the *Arabidopsis*, apple, rice, and maize genome files were downloaded from the NCBI online website. First, a hidden Markov model (HMM) search against the rose genome database was conducted with the DNA domain using TBtools-II (Toolbox for Biologists) v2.136. All putative candidate members were assessed using Pfam [41] (https://www.ebi.ac.uk/interpro/, accessed on 18 June 2024) and NCBI-CDD [42] (https://www.ncbi.nlm.nih.gov/cdd, accessed on 20 June 2024). Second, to further determine whether the identified proteins belonged to the *CAMTA* gene family, we analyzed the protein structural domains using SMART (https://smart.embl.de, accessed on 23 June 2024). Finally, we obtained members of the *R*. *chinensis CAMTA* gene family and analyzed the physicochemical properties of these genes using EXPASY (https://web.expasy.org/protparam/, accessed on 25 June 2024). Prediction of the subcellular localization of proteins was carried out using the online software Plant-mPLoc (v2.0) (http://www.csbio.sjtu.edu.cn/bioinf/plant-multi/, accessed on 28 June 2024). The gene structure of each member of *CAMTA* was analyzed using the “Visualize Gene Structure (from GTF/GFF3 File)” function in TBtools-II software.

### 4.2. Secondary and Tertiary Structure Prediction of Rosa Chinensis CAMTA Protein

The secondary structure of the *CAMTA* gene-encoded protein in *R*. *chinensis* was predicted using the online software SOPMA [43] (https://npsa-prabi.ibcp.fr/, accessed on 5 July 2024), The data and images were exported, which included four different forms of composition: α-helical, β-turned-angle, irregularly coiled, and chain-extended structures. The tertiary structure of the *R. chinensis* CAMTA protein was predicted via the website SWISS-MODEL (https://swissmodel.expasy.org/interactive, accessed on 5 July 2024). The protein interaction network of the rose CAMTA family was constructed using STRING 12.0 (https://cn.string-db.org/, accessed on 10 July 2024). The selected value was >0.400, and the interaction prediction analysis was carried out to investigate the roles of the *R*. *chinensis* CAMTA protein members.

### 4.3. Sequence Comparison and Phylogenetic Analysis of Rosa Chinensis CAMTA Genes

The *RcCAMTA* sequence was compared and analyzed using Jalview software (https://www.jalview.org/, accessed on 23 July 2024). Multiple sequence alignment analysis revealed that members of the *RcCAMTA* family have similar conserved functional domains. Phylogenetic analyses were performed using MEGA11, and evolutionary trees were constructed using the maximum likelihood method. The analysis included 5 rose CAMTA proteins, 6 Arabidopsis CAMTA proteins, 18 rice CAMTA proteins, and 7 maize CAMTA proteins. The parameters were set as follows: the boot replication value was set to 1000, and the other parameters were set as in [44]. In addition, the evolutionary tree was visualized and enhanced using the Evolview website (https://evolgenius.info//evolview-v2/#login, accessed on 17 July 2024).

### 4.4. Gene Structure and Conserved Motif Analysis of Rosa Chinensis CAMTA Genes

The conserved motif positions were predicted using the online software MEME [45] (http://meme-suite.org/tools/meme, accessed on 23 July 2024), A maximum of 10 motifs were searched for, with the other parameters remaining at their default settings. Then, the corresponding motif information and the evolutionary tree information of the *R*. *chinensis CAMTA* family derived from MEGA11 were combined to visualize the conserved motifs of the *R*. *chinensis CAMTA* members via the “gene structure view (Advances)” function of TBtools-II software.

### 4.5. Collinearity Analysis of Rosa Chinensis CAMTA Genes

The online database ensemble was used to determine the genetic relationship between *A. thaliana* L. and *R*. *chinensis* and between *R. chinensis* and *Malus pumila*. The genome files (FASTA) and genomic annotation file (GFF3) of the related *R. chinensis CAMTA* gene family were downloaded, and the collinearity analysis was performed using the “Text Merge for MCScanX” function in TBtools software [46]. Finally, the results were visualized using the “Multiple Systeny Plot” function in TBtools software.

### 4.6. Cis-Acting Element Analysis of Rosa Chinensis CAMTA Genes

The “Gtf/Gff3 Sequences Extract” and “Fasta Extract (Recommended)” functions of TBtools were used to extract the 2000 upstream *CAMTA* genes of rose from the *R. chinensis* gene databases [47]. The PlantCARE database was used to search for the cis-regulatory elements in the *R. chinensis* gene promoter region to study the role of genes in hormonal response.

### 4.7. Plant Material and Treatments

*R. chinensis* “Movie Star” was used as the experimental material in this study. Cut rose flowers were evenly soaked in distilled water for 2 h and then trimmed to about 30 cm long in water, with 2–3 compound leaves kept intact. The cut rose flowers were evenly soaked in distilled water and cultured for 14 d at an ambient temperature of 25 °C. Fresh cut flowers that grew evenly were selected and treated with 500 mL of 50 μM SA [48], 10 mg/L IAA [49], 0.5 μmol/L ABA [50], or 5 mg/L MeJA [51]. The samples were collected at 6, 12, 24, 48, and 96 h after treatment [52]. The control was an equal volume of ultrapure water treatment, which was changed every day. The flower diameter was recorded every day after treatment.

### 4.8. RNA Isolation and qRT-PCR Analysis

To analyze the spatial expression patterns of the *RcCAMTA* genes in rose, samples (0.3 g) were collected from the stems, leaves, petals, pistil, stamen, calyces, receptacles, and thorns of the cut roses. The samples were then immediately frozen and stored at −80 °C.

Total RNA was extracted from the samples using TRIzol reagent (Invitrogen, Carlsbad, CA, USA). The purity and concentration of the RNA were then examined using a Pultton P100+ ultra-micro spectrophotometer (Wuzhou Dongfang, Beijing, China). RNA samples with A260/A280 ratios between 1.9 and 2.1 were chosen for the subsequent experiments. Then, the Fast Quant First Strand cDNA Synthesis Kit (Tianen, Beijing, China) was used to synthesize the cDNA. These reactions were executed under the following conditions: 37 °C for 15 min, 85 °C for 5 s, and then terminated at 4 °C. The SYBR Green Premix Pro Taq HS Premix kit (Accurate Biotechnology (Hunan) Co., Ltd) was used for qRT-PCR with a LightCycler 480 Real-Time PCR System (Roche Applied Science, Penzberg, Germany). *RcACTIN* (AB239794) was used as an internal reference gene, and the reaction system consisted of 2 µL of cDNA, 0.4 µL of primer F, 0.4 µL of primer R, 10 mL of 2× SYBR GreenPro Taq HS Premix (Accurate Biotechnology (Hunan) Co., Ltd), and 7.2 µL of RNase-free ddH_2_O. The primers used in qRT-PCR were designed with Sangon Biotech (Shenggong Biological Engineering Co., Ltd). The specificity of the primers was determined through melt curve analysis. PCR cycling was performed as follows: 2 min at 95 °C followed by 39 rounds of 5 s at 95 °C, 30 s at the optimal annealing temperature, and finally 1 cycle of 5 s at 65 °C. A melting curve (65–95 °C at increments of 0.5 °C) was generated to verify the specificity of primer amplification. Three replicates of each tissue sample were used to account for possible sampling and experimental errors. The sequences of the above primers are listed in Appendix A. The data were analyzed using the 2^-−∆Ct^ calculation method [53]. All experiments in our study were repeated three times independently.

### 4.9. Statistical Analysis

Analysis of the data was conducted using SPSS version 22.0. Three biological replicates were used in all experiments. All data were gathered from three independent experiments. One-way ANOVA was used to analyze statistical differences among the measurements taken at different times or under different treatment conditions.

## 5. Conclusions

This study identified five *RcCAMTA* genes from the rose reference genome, which are distributed on three chromosomes. Then, the gene structure, gene localization, protein interactions, conserved motifs, phylogenetic relationships, collinearity, and cis-acting elements of the five *RcCAMTA* genes were analyzed. According to the expression levels of the *RcCAMTAs*, the rose *CAMTA* family genes have different spatial expression patterns in different tissues. Under different hormone treatments (SA, IAA, ABA, and MeJA), it was found that all *RcCAMTA* members can be induced by one or more treatments. Among them, *RcCAMTA2* was significantly induced under the IAA and MeJA treatments, while *RcCAMTA3* was significantly induced under the ABA treatment. All *RcCAMTA* genes significantly responded to SA treatment, but the expression patterns were different. Thus, *RcCAMTAs* play an important role in hormone signaling transduction in rose. These results lay the foundation for further research on the potential functions of members of the rose *CAMTA* family.

## Figures and Tables

**Figure 1 plants-14-00070-f001:**
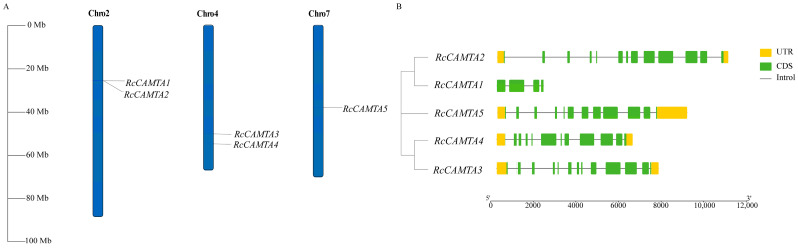
The distribution of *CAMTA* gene family members of chromosomes and structural analysis in rose. (**A**) Chromosome localization analysis. Each vertical line represents the number of chromosomes, with gene names conveniently displayed at the top of their respective chromosomes. (**B**) Structure analysis of the *CAMTA* gene in rose.

**Figure 2 plants-14-00070-f002:**
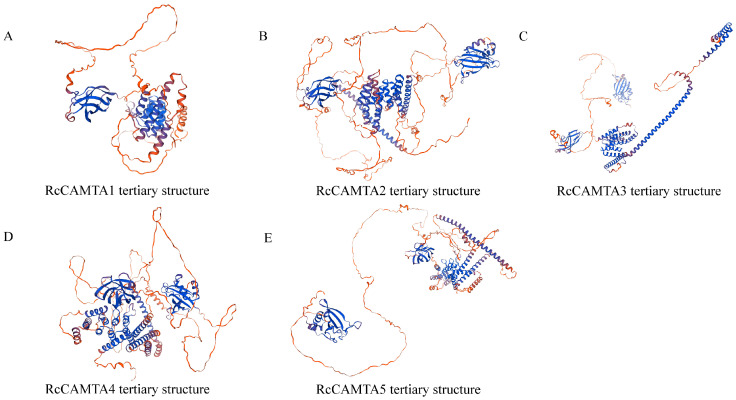
Predicted tertiary structure of RcCAMTAs. (**A**) RcCAMTA1 tertiary structure. (**B**) RcCAMTA2 tertiary structure. (**C**) RcCAMTA3 tertiary structure. (**D**) RcCAMTA4 tertiary structure. (**E**) RcCAMTA5 tertiary structure. Note: Blue color indicates α-helix; purple color indicates chain-extended structure; red color indicates irregular coiling.

**Figure 3 plants-14-00070-f003:**
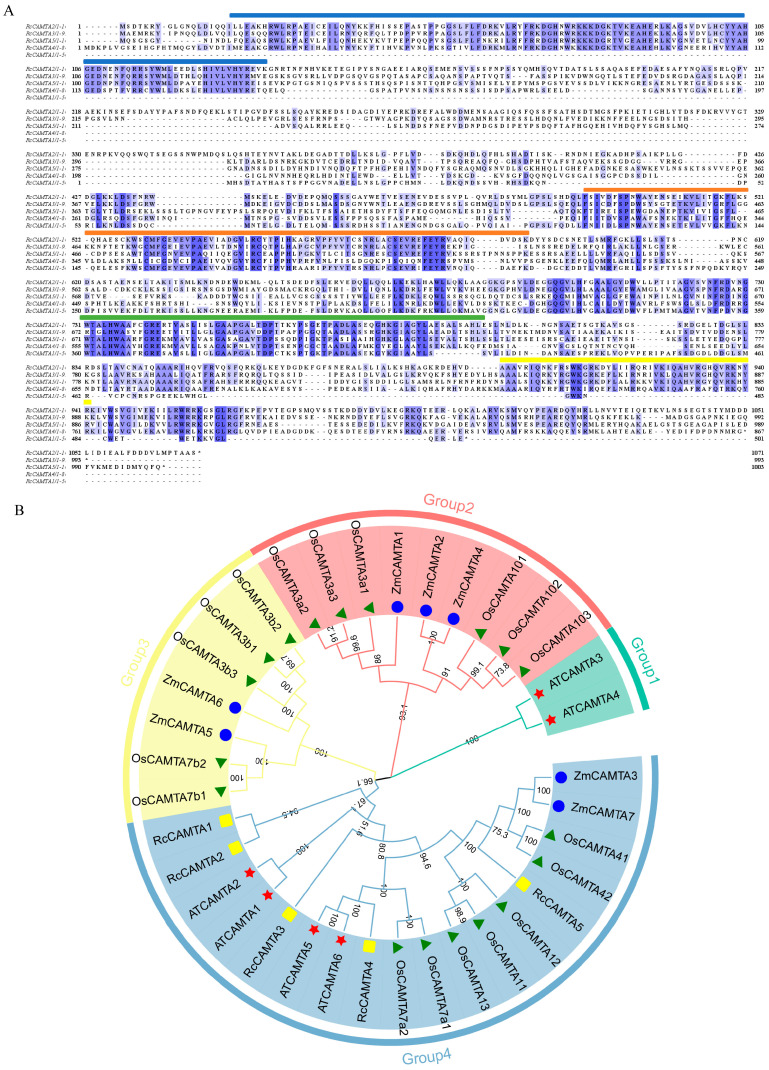
Multiple sequence alignment and phylogenetic tree of *RcCAMTAs*. (**A**) Conserved structural domains of the rose *CAMTA* gene family. Blue boxes are CG-1 DNA-binding structural domains, orange boxes are IPT structural domains, green boxes are ANK repeat structural domains, and yellow boxes are IQ motifs. (**B**) Phylogenetic tree of *RcCAMTAs*. The four different colored shapes represent CAMTA proteins from four different species. Yellow rectangles, red stars, green triangles, and blue circles represent CAMTA proteins from rose, Arabidopsis, rice, and maize, respectively.

**Figure 4 plants-14-00070-f004:**
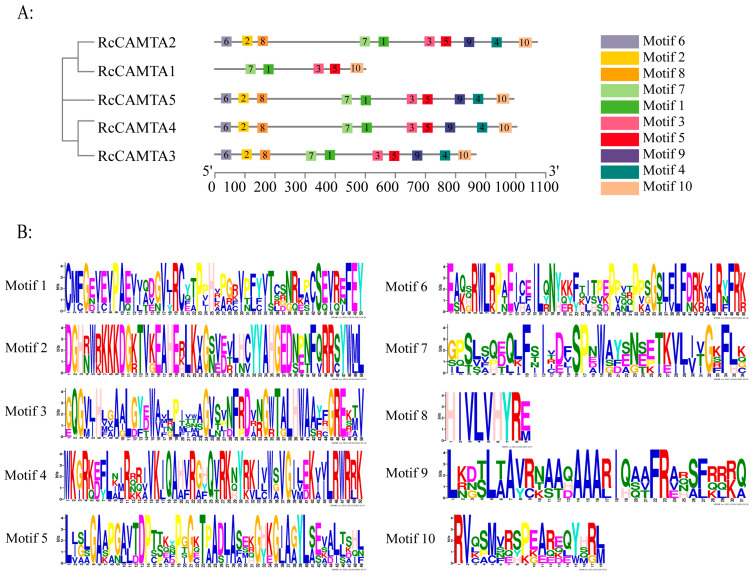
The distribution and composition of CAMTA proteins in rose pertaining to motif occurrence. (**A**) Distinct motifs are represented by colored boxes. (**B**) Sequential stacks of letters illustrate the amino acid sequences associated with each motif. The overall height of the stack signifies the information content in bits of the respective amino acid at each position of the motif.

**Figure 5 plants-14-00070-f005:**
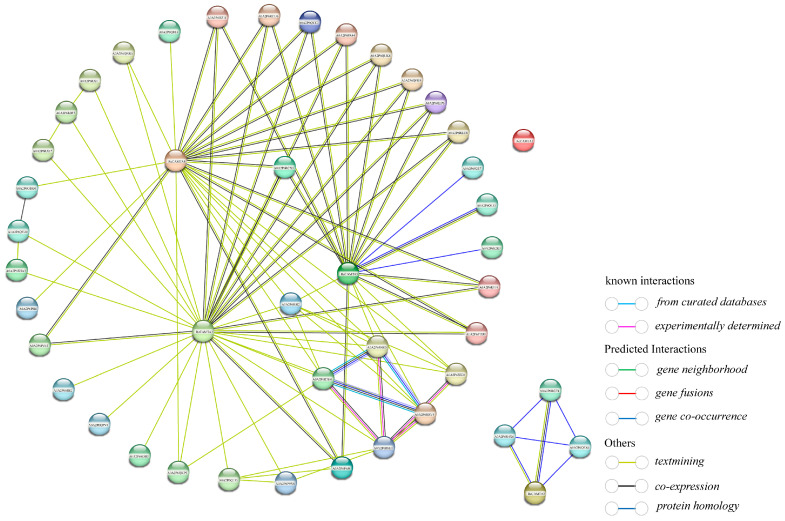
Interaction network of rose CAMTA proteins.

**Figure 6 plants-14-00070-f006:**
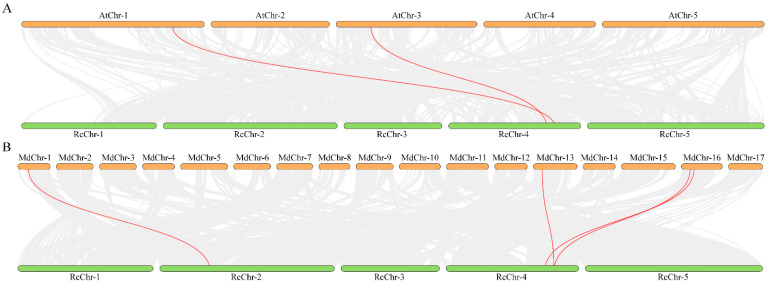
Collinearity analysis of *CAMTA* family genes rose (**A**) *R. chinensis* and Arabidopsis gene collinearity analysis, *RcChr*: *R. chinensis* chromosome, *AtChr*: *A. thaliana* L. chromosome; (**B**) *R. chinensis* and *apple* gene collinearity analysis, *RcChr*: *R. chinensis* chromosome, *MdChr*: apple chromosome. The gray lines indicate duplicated blocks, while the red lines indicate duplicated *RcCAMTA* gene pairs.

**Figure 7 plants-14-00070-f007:**
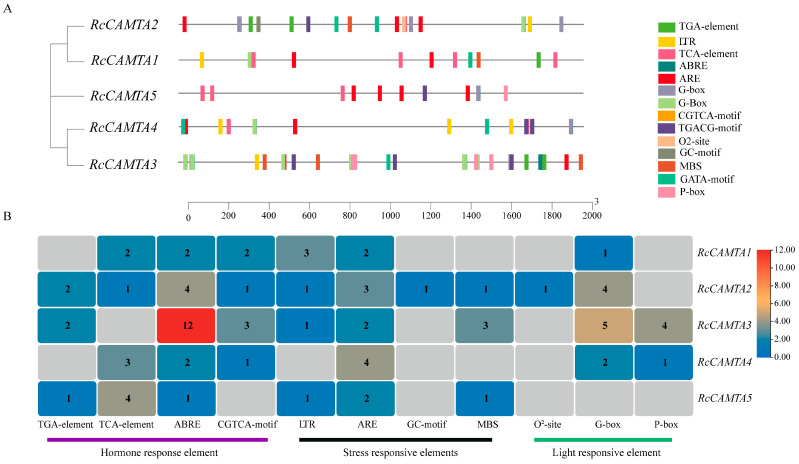
*RcCAMTA* cis-acting element analysis. (**A**) Rose *CAMTA* cis-acting element. (**B**) Analysis of the number of cis-acting elements of the *CAMTA* gene.

**Figure 8 plants-14-00070-f008:**
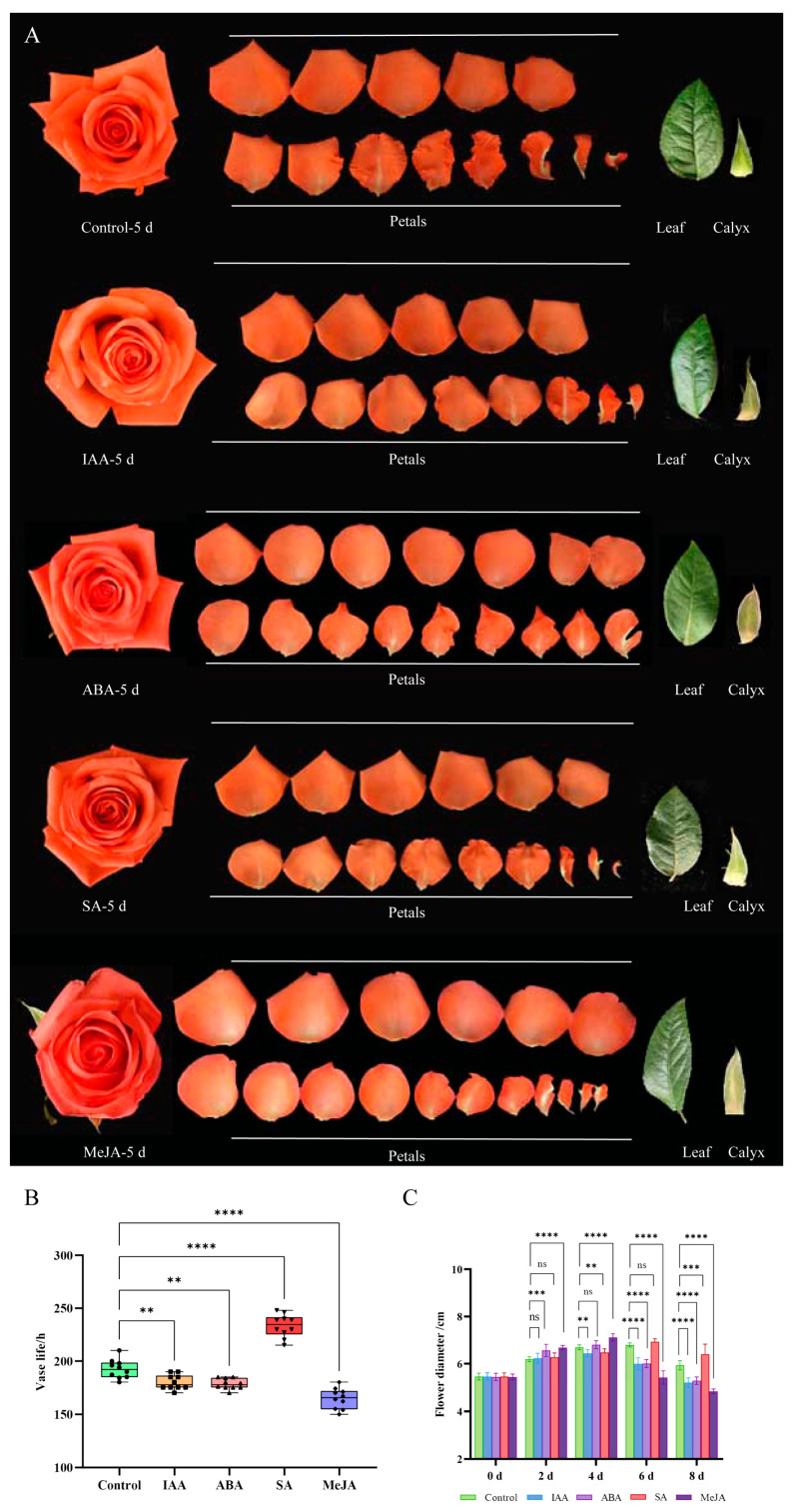
The effect of phytohormones on the preservation of cut roses after harvesting. (**A**) Schematic diagram of rose under different phytohormone treatments (IAA, ABA, SA, and MeJA) for 5 days. (**B**) Postharvest vase life of roses under different phytohormone treatments. (**C**) Flower diameter of roses under different phytohormone treatments (ns < 0.1234, * *p* < 0.0332, ** *p* < 0.0021, *** *p* < 0.0002, and **** *p* < 0.0001, one-way ANOVA, Tukey test).

**Figure 9 plants-14-00070-f009:**
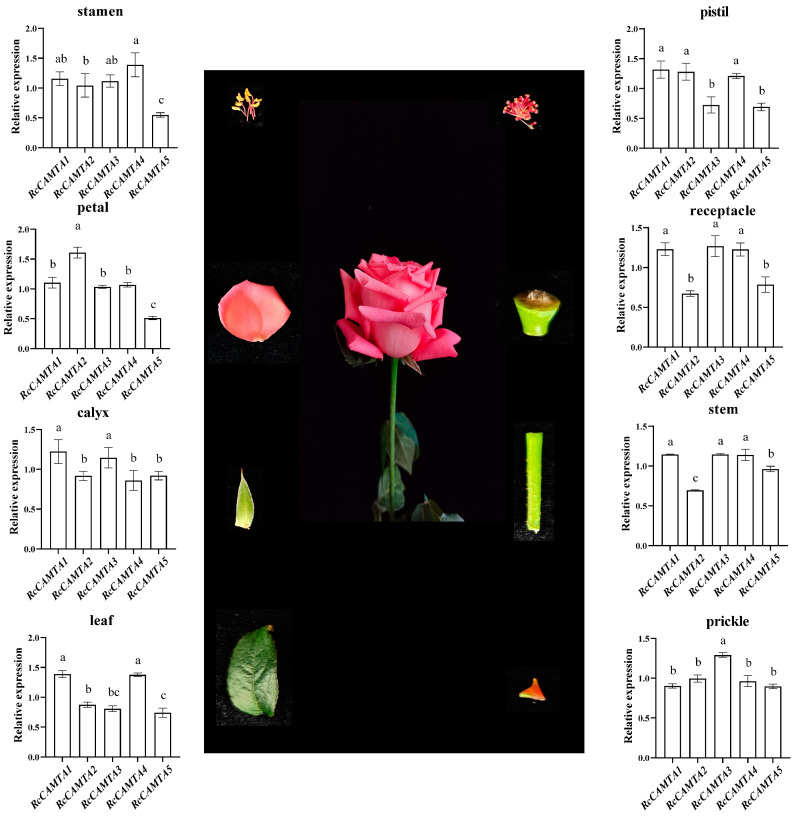
Tissue-specific expression analysis of 5 *RcCAMTAs*. The error bars signify the standard error, which was calculated based on three independent replicates. The relative expression of each gene in distinct tissues is presented as the mean ± SE (*n* = 3). Bars labeled with different lowercase letters indicate significant differences, as determined by Duncan’s multiple range tests (*p* < 0.05).

**Figure 10 plants-14-00070-f010:**
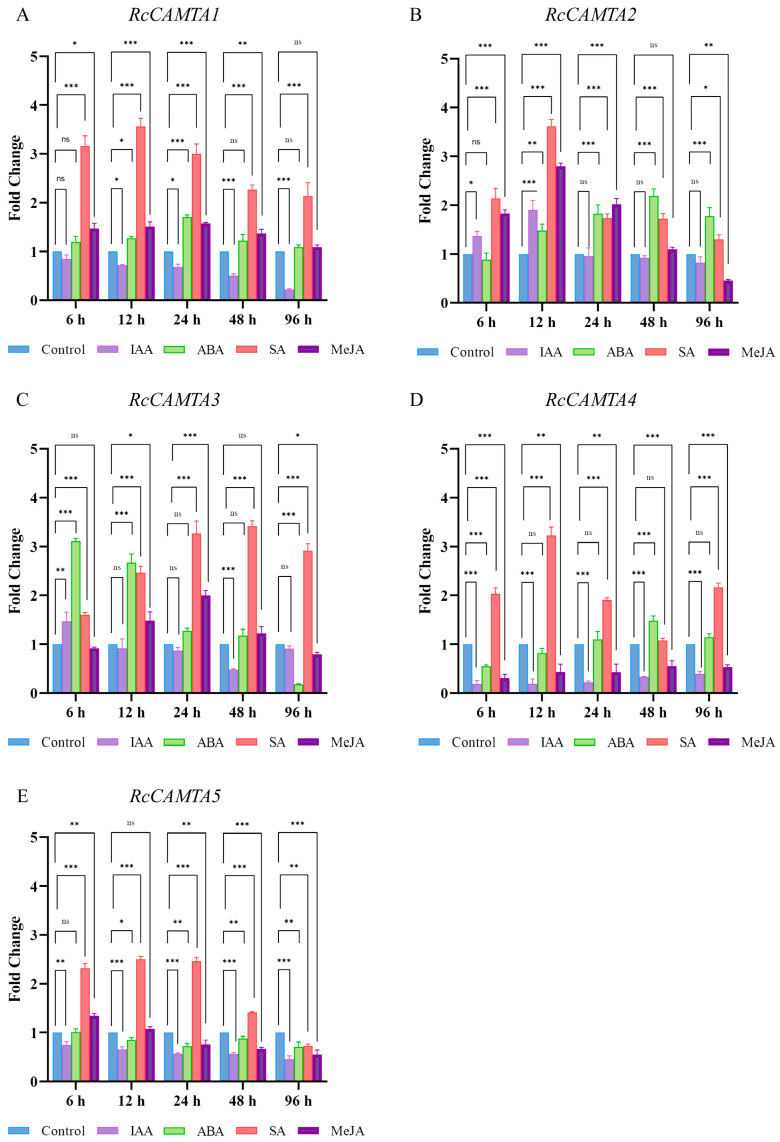
Expression fold changes of *CAMTA* genes were analyzed in the petal of rose. The expression patterns of *RcCAMTA1* and *RcCAMTA5* genes at different times of different treatments are shown in (**A**–**E**). The asterisk (*) indicates the corresponding gene expression that was significantly up- or down-regulated compared with the control (ns < 0.012, * *p* < 0.033, ** *p* < 0.02, and *** *p* < 0.001, one-way ANOVA, Tukey test).

**Table 1 plants-14-00070-t001:** Physicochemical properties of the *RcCAMTAs*.

Gene	Gene ID	Chr. No	Chr. Location	Length (aa)	Mol. Wt. (KDa)	pI	Instability Index	Aliphatic Index	Grand Average of Hydropathicity	Subcellular Localization
*RcCAMTA1*	RchiOBHm_Chr2g0113251	2	25,422,276–25,424,462	500	55.5316	5.04	40.26	80.14	−0.384	Endoplasmic reticulum Nucleus
*RcCAMTA2*	RchiOBHm_Chr2g0113351	2	25,531,386–25,542,220	1070	120.25298	5.46	42.42	74.03	−0.602	Nucleus
*RcCAMTA3*	RchiOBHm_Chr4g0424431	4	50,049,131–50,056,720	866	98.17811	7.65	44.27	81.67	−0.493	Nucleus
*RcCAMTA4*	RchiOBHm_Chr4g0429961	4	54,664,331–54,670,699	1002	112.18003	5.58	49.11	75.90	−0.539	Nucleus
*RcCAMTA5*	RchiOBHm_Chr7g0219321	7	37,709,032–37,717,932	992	110.86922	8.54	41.47	77.39	−0.531	Nucleus

Note: pI, isoelectric point. Mol. Wt., molecular weight.

**Table 2 plants-14-00070-t002:** Predicted secondary structure of RcCAMTAs.

Protein	Alpha Helix (%)	ExtendedStrand (%)	Beta Turn (%)	Randon Coil (%)	Distribution of Secondary Structure Elements
RcCAMTA1	38.80	5.40	0.00	55.80	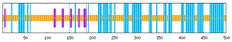
RcCAMTA2	32.80	4.21	0.00	62.99	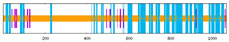
RcCAMTA3	44.34	5.43	0.00	50.23	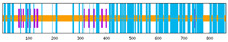
RcCAMTA4	35.53	4.29	0.00	60.18	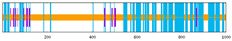
RcCAMTA5	40.62	3.33	0.00	56.05	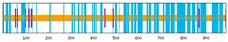

Note: Blue color indicates α-helix; purple color indicates chain-extended structure; orange color indicates irregular coiling.

## Data Availability

Data are contained within the article and Appendix A.

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
