# Peer review of "Genome-Wide Identification and Expression Analysis of the CAMTA Gene Family in Roses (Rosa chinensis Jacq.)"

_plants, 2024, doi:10.3390/plants14010070_

Round 1
Reviewer 1 Report
Comments and Suggestions for Authors
The manuscript addresses the in silico analysis of CAMTA gene family in Rosa chinensis and studies the expression pattern of the identified genes in different tissues and under different hormone and signaling molecules. While the approach is predominantly hypothetical, it explores aspects that may prove useful for future researchers conducting wet lab experiments. The manuscript needs significant revision before considering for publication. I have listed my comments below:
Comment 1: (P4,L116-118) How the analysis was done, which software were used must be moved to 'Materials and Methods' section. Please make corrections for the whole 'Result' section.
Comment 2: (P5,L137-139) Should be moved to the 'Materials and Methods' section.
Comment 3: (P7,L158) Please rewrite this sentence.
Comment 4: (P7,L173) Please move Table 3 in supp files.
Comment 5: (P10,L238) Please move table 4 into supp file.
Comment 6: (P13,L241) Divide the Figure 7 into two figures. Add A,B, and F in one figure and E, C, and D in one figure.
Comment 7: (P14,L267) When fold change value is used for showing gene expression, it is recommended to use ‘expression change’ rather than ‘expression level’. Please correct accordingly.
Comment 8: (P16,L298) Figure 8 shows the expression level whereas Figure 9 shows the expression change relative to control. It is recommended to use 'Expression fold change' instead of 'Relative expression' here.
Comment 9: (P16,L301) The authors need to justify why they use 0hr control for all the time-points? For a few hours difference it may be fine but for 24h, 48h, and 96h using 0h samples as control is not justified, especially when they are using cut-flowers.
Comment 10: (P17,L320) Please perform gene duplication analysis.
Comment 11: (P17,L332)Please write the number of genes directly.
Comment 12: (P17,L336) During citing figures, please also cite the panel (A,B..).
Comment 13: (P18,L376) Please move Table 5 in the supp files.
Comment 14: (P18,L400) Please move this sub-section after 4.7.
Comment 15: (P19,L411) Change to 'From the NCBI'. Please look for similar mistakes and correct them.
Comment 16: (P19,L428-429) Delete the repeated sentence.
Comment 17: (P19,L432) Please italicize when you mean CAMT gene. Please make corrections throughout the manuscript.
Comment 18: (P19,L436) Only write full genus name at the first mention counting from the 'Introduction'. Please make corrections throughout the manuscript.
Comment 19: (P19,L436) 'CAMTA' must not be italicized when referred to as protein. Please make corrections throughout the manuscript.
Comment 20: (P19,L443) It is recommended to write "RcCAMTA" instead of "CAMTA sequence of Rosa chinensis".
Comment 21: (P19,L444-446) Please write which plant species were included in the phylogenetic analysis and how many proteins from each species was included.
Comment 22: (P20,L458) Why not include maize and rice as well. Especially when they were used in gene identification.
Comment 23: (P20,L488) Should be “independently”.
Author Response
Dear Editor,
Thanks a lot for having reviewed our manuscript (plants-3338324). We have revised the manuscript, and would like to submit it for your consideration. According to your comments and suggestions, we have made corresponding changes. The revisions have been highlighted in the revised manuscript.
I greatly appreciate both your help and that of the referees concerning improvement to this paper. Below you can find point-to-point responses to Reviewers’ comments. We hope that the revised version of the manuscript is now acceptable for publication in your journal.
I look forward to hearing from you soon.
We would like to express our sincere thanks again to you for the constructive and positive comments.
With best wishes,
Yours sincerely,
Wanyi Su; Weibiao Liao

Reviewer 2 Report
Comments and Suggestions for Authors
Line 45 should this read NLS, not NlS? Also you should define NLS and TLG domains please.
A few clarifications are needed line 94 which gene is RcCAMTA2-5 or is this genes 2 through 5. Also how can the gene only have 2 introns if they have 4-13 exons introns should be n-1. Please clarify.
Most proteins are listed as kDA not Da. please correct this.
In figure 7E can you please define normal versus abnormal petals. This does not appear to be in the text.
DO you think CAMTA1 is a pseudogene and not a functional member of the family? It is half the size and appears to lack many of domains you talk about and has no interaction partners and possibly located to the ER. Can you add to the discussion a little to say if any other know CAMTAs are short like this or are most closer to 850-1100 amino acids.
Also please highlight the NLS motifs in figure 3A if possible.
Author Response

(The authors gave the same response as above.)

Reviewer 3 Report
Comments and Suggestions for Authors
The Authors presented in the manuscript interesting results regarding genome-wide Identification and expression analysis of calmodulin-binding transcription activator (CAMTA) gene family in rose (Rosa chinensis Jacq.). Among which the most important results are the following:
Five rose CAMTA genes were identified. RcCAMTA gene members were located on chromosomes 2, 4 and 7. CAMTA genes are classified into three subfamilies. The cis-acting element prediction results showed that the rose CAMTA gene family contains phytohormone signaling response elements, abiotic stress response, light response and other elements, most of which are hormone signaling response elements. qRT-PCR analysis showed that all 5 rose CAMTAs responded to salicylic acid (SA). RcCAMTA3 was significantly induced by abscisic acid (ABA), RcCAMTA2 was induced by 1H-Indole-3-acetic acid (IAA) and methyl jasmonate (MeJA).
In my opinion, the manuscript contains important research findings, and it may be considered for publication, after including the following minor revisions:
- The Authors stated that “The samples were collected at 6, 12, 24, 48 and 96 h after treatments” (line 406) – it should be explained on what basis these time-points were selected during the investigations.
- “4.8. RNA Isolation and qRT-PCR Analysis” (line 470) – in this section, it should be added information about the post-PCR analyses of the amplicons (i.e. melting curve analyses). The Authors used SYBR Green in the gene expression studies; therefore, it is highly recommended to evaluate if there were no-specific amplicons in the PCR products .The graphs evidencing the lack of additional non-specific amplification should be placed in the Supplementary File.
- In Materials and Methods, the subchapter regarding Statistical analyses should be added.
- Minor/moderate revision of the English style and grammar should be conducted by the native speaker - specialist in the molecular biology.
Author Response

(The authors gave the same response as above.)

Round 2
Reviewer 1 Report
Comments and Suggestions for Authors
Much improvement has been observed. For figure 10, please change the labels
of the figures to 'Fold change' instead of 'Relative expression'.
Author Response

(The authors gave the same response as above.)
